# Understanding frailty among older people living in old age homes and the community in Nepal: A cross-sectional study

**Richa Shah**[1,2] *, **Rogie Royce Carandang**[2], **Akira Shibanuma**[2], **Ken Ing Cherng Ong**[2], **Junko Kiriya**[2], **Masamine Jimba**[2]

**1** Health Action and Research, Kathmandu, Nepal, **2** Department of Community and Global Health, Graduate School of Medicine, The University of Tokyo, Tokyo, Japan

* richa.np@gmail.com

**Data Availability Statement:** The data is available at https://doi.org/10.6084/m9.figshare.14158559.v1.

## Abstract

### Introduction

Frailty is a state of being vulnerable to adverse health outcomes such as falls, delirium, and disability in older people. Identifying frailty is important in a low-income setting to prevent it from progressing, reducing healthcare costs, increasing the chances of reversibility, and implementing effective interventions. The factors affecting frailty in older people living in old age homes could differ from those living in the community. This study was conducted to identify the factors associated with frailty in older people residing in old age homes and communities in Kathmandu Valley, Nepal.

### Methods

This is a cross-sectional study conducted from April to June 2019 in three districts of Kathmandu Valley, Nepal. Data were collected from 193 older people residing in old age homes and 501 residing in communities aged 60 and above using convenience sampling. Frailty was measured using the Groningen Frailty Indicator. Data were collected via face-to-face interviews. Multiple linear regression analyses were used to examine the association between independent variables and frailty.

### Results

Frailty was more prevalent among older people in old age homes (71.5%) compared to those in the community (56.3%). Older people who were satisfied with their living environment had lower frailty scores in both old age homes ($\beta = -0.20$, $p<0.01$) and the community ($\beta = -0.15$, $p<0.001$). Those who had self-rated unhealthy lifestyle had higher frailty scores in both old age homes ($\beta = 0.45$, $p<0.001$) and the community ($\beta = 0.25$, $p<0.001$). In the community, those over 80 years of age had higher frailty scores ($\beta = 0.15$, $p<0.01$) and those with higher education had lower scores ($\beta = -0.13$, $p<0.05$).

**Funding:** The author(s) received no specific funding for this work.

**Competing interests:** The authors have declared that no competing interests exist.

## Conclusion

The living environment and lifestyle are key modifiable risk factors of frailty, both in old age homes and the community. The findings suggest a need for lifestyle modification and reforms in building standards, especially in old age homes, to promote age-friendly communities.

## Introduction

The aging population has been increasing worldwide as a result of declining fertility, improved health, and decreased mortality [1]. The number of people aged 60 and above is expected to increase from 962 million in 2017 to 2.1 billion in 2050, globally [2]. The transition to an aging society is faster in low- and middle-countries (LMICs) compared to that in high-income countries [3]. This increase is so rapid and sudden that most countries are not prepared to deal with the associated challenges of aging, such as frailty [3].

Frailty is defined as a state of being vulnerable to adverse health outcomes such as falls, delirium, and disability [4, 5]. It is a multidimensional concept [6] encompassing physical, social, cognitive, and psychological aspects [7]. Old age leads to a decline in physiological and functional reserve capacity of multiple systems and frailty occurs when this reserve capacity is critically low [4, 8]. Although frailty is common in old age, it is not a manifestation of old age [4, 9] and is reversible with appropriate interventions [8]. The factors associated with frailty are age, income, multimorbidity, female gender, alcohol consumption, and smoking [9, 10]. Progression of frailty can be dealt with resistance exercises, addition of nutritional supplements in the diet, and cognitive training [11–14]. Frailty is used as a predictor for older people's mortality and disability [15] and may discover unrecognized health problems [16, 17].

Frailty is likely to increase globally, including in LMICs such as Nepal, which poses a public health challenge. The prevalence of frailty is higher in upper-middle-income countries compared to high-income countries. Since most studies on frailty were conducted in high- and upper-middle-income countries, the evidence from low- and low-middle income countries remains scarce [18–21]. Due to a lack of evidence, health and social care planning in these countries are difficult [22]. In Nepal, for instance, little is known about the burden of frailty and the factors leading to frailty in older people.

The number of older people in Nepal is rising rapidly as elsewhere in the world [23, 24]. The Government of Nepal defined people who completed 60 years of age as senior citizens [25]. However, this manuscript uses the term older people instead of senior citizens as the term older people is less discriminatory and biased and hence, more appropriate to denote people aged 60 and over [26]. According to the latest census in 2011, the population of individuals aged 60 years old and above was 2.1 million in Nepal, which was 8.1% of the total population [27]. The average life expectancy in Nepal was 70.2 years in 2018 compared to 27.0 years in 1951 and 64.0 years in 2008, which is a drastic improvement [28].

Although most Nepali older people live at home, the number of older people living in old age homes is increasing because of a lack of support from their children [24, 29]. The major causes are urbanization, preference for nuclear families, and migration of adults in their prime age to urban areas and abroad for better opportunities [29]. These changes have made older people to live alone, care for their own needs, and look for an alternative living arrangement such as old age homes. Although the concept of old age home is not well-established in Nepal, it generally refers to shelter or multi-residence housing facility for older people who are

helpless or do not have children to take care of them [23, 30]. These facilities provide residence, meals, gatherings, recreation activities, and some form of health care [30]. The rising number of old age homes and their residents calls attention to their better health, especially issues involving frailty, along with older people residing in the community.

Identifying frailty is important in a low-income setting with limited healthcare resources because it identifies people who need additional medical care. Early identification of frailty is essential to prevent it from progressing, sourcing healthcare to those in need, reducing healthcare costs, increasing the chances of reversibility, implementing effective interventions, and preventing adverse health outcomes such as disability [31, 32]. The factors affecting frailty in older people living in old age homes could be different from those living in the community. However, most studies have focused on frailty status in either the old age homes or the community [5, 19, 20]. This study was conducted to identify the factors that affect frailty status in community-dwelling older people and those living in old age homes in Kathmandu Valley, Nepal.

## Materials and methods

### Study design and settings

This was a cross-sectional study conducted in old age homes and communities of Kathmandu valley in Nepal. Kathmandu valley is comprised of three districts, namely Kathmandu, Lalitpur, and Bhaktapur. Kathmandu, the capital city of Nepal, and adjacent districts, which are Lalitpur and Bhaktapur are densely populated. These districts have a higher number of old age homes compared to the rest of the country. The number of old age homes amounted to 82 all over Nepal in 2012 and about 1500 older people resided there [33]. An updated and official list of old age homes in Nepal is unavailable. However, based on the principal investigator's (RS) search, 21 old age homes were in Kathmandu Valley, which provided residence to about 350 older people.

### Sample size and sampling procedure

The sample size was calculated based on a previous study by Lin et al., which divided the population into two groups, frail and robust with means of 52.6 (SD [Standard deviation] 8.8) for frail and 56.2 (SD 12.8) for non-frail [34]. OpenEpi version 3.01 was used for calculation with 80% power, 5% significance level, and 95% confidence interval. Thus, the minimum sample size obtained was 115 for old age homes and 348 for the community. However, anticipating a 30% refusal rate and incomplete responses, the final minimum sample size was set to 164 for old age homes and 497 for the community.

Convenience sampling was used to select the older people in the community as data on the number of older people aged only over 70 was available. The heads of 20 wards offices from Kathmandu, six from Lalitpur, and five from Bhaktapur districts were approached with a request letter for permission to conduct the study from The University of Tokyo. These wards were selected purposively based on the ease of access. A ward is the smallest local unit or political division used for electoral purposes [35]. In Kathmandu district, verbal permission was obtained from six wards, in Lalitpur district from three wards, and three wards in Bhaktapur district. Local leaders from the selected wards assisted in approaching older people. Older people were recruited through home visits based on the inclusion and exclusion criteria.

An official list of old age homes in Nepal was not available. A non-governmental organization called Ageing Nepal, which works for the welfare of the older people in Nepal, provided its own list of old age homes in Nepal. The heads of 15 old age homes were approached first via telephone calls followed by face-to-face meetings with a request letter for permission from

The University of Tokyo. Among them, ten provided verbal permission, after which interviews were conducted.

## Participants

The older people included in this study were aged 60 years and over and residing in the community and old age homes in Kathmandu Valley at the time of data collection [23, 30]. At old age homes, medical records were available. Based on the records, older people were excluded from this study if they suffered from moderate to severe cognitive impairment, severe hearing impairment, muteness, and mental illnesses such as psychosis, bipolar disorder, and schizophrenia. Most community-dwelling older people also had their medical records which the interviewers could access. In the absence of medical records or inadequate information in them, the older people were excluded from the study if they had visible signs of cognitive impairment such as inability to remember their name or the location of their residence, which affected the ability to comprehend the study procedure and to cooperate with the investigators. Sensory and cognitive decline may occur in old age, resulting in loss of memory, impaired comprehension and judgment, or inability to respond accurately. They could incapacitate older people from consenting to participate in the study or providing precise information. Only one older person was recruited from each household based on inclusion and exclusion criteria in the community. If a couple resided together in an old age home, only one of them was recruited. Based on the above, 35 older people were excluded from this study.

## Variables and assessment

The framework for this study was adapted from a study in the Netherlands conducted by Gobbens et al. (Fig 1) [6]. It outlines the factors leading to frailty and the stage at which health promotion and prevention activities can be undertaken to delay frailty.

**Frailty.** The outcome variable was frailty. Groningen Frailty Indicator (GFI) was used to assess frailty, which is a validated instrument and tested in multiple settings [7, 8, 36]. It has 15 items that assess four domains: physical (nine items), cognitive (one item), social (three items), and psychological (two items) [7]. The responses were obtained in three categories (yes, no,

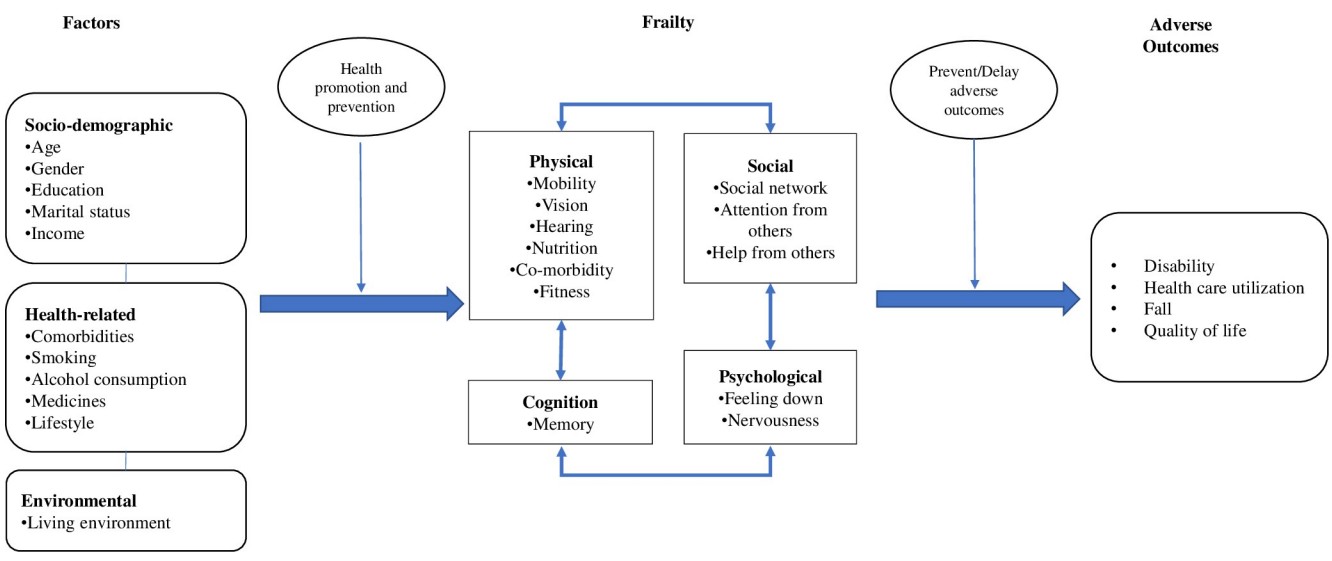

**Fig 1. Conceptual framework.**

and sometimes) and dichotomized as 0 and 1 based on the guidelines of GFI [7]. The responses were dichotomized as "yes" (0) and "no" (1) for grocery shopping, walk outside house, getting dressed, and visiting restroom. For rest of the questions, the responses were "yes" (1) and "no" (0). For the question on cognition, the response "sometimes" was coded as "0" whereas for other questions, "sometimes" was coded as "1". The maximum possible score was 15 and the lowest possible score was 0 [7]. A score of 4 or higher is considered as "moderate" or "severe" frailty [7, 37]. The internal consistency of GFI in this study was measured using Cronbach's alpha which was 0.74 for old age homes, 0.76 for the community, and 0.75 for combined, indicating high reliability [38].

**Socio-demographic and health characteristics.** The socio-demographic variables included age, gender, marital status, education level, income, and satisfaction with the home living environment. These variables were selected based on past research articles [6, 7]. Age was recorded in completed years. Health-related variables were the presence of comorbidities, smoking (currently smoking, never, and past smoker), alcohol consumption (currently consuming, never, and in the past), number of prescribed medicines taken in a day, and self-rated health (healthy, fair, and unhealthy). Marital status was classified into single (divorced, unmarried, widow, or widower) and married (married or cohabitation).

## Data collection

An interviewer-administered questionnaire (tablet-based) was used for face-to-face interviews. The questionnaire was initially prepared in English and translated into the Nepali language by a bilingual medical doctor and public health professional. Two independent bilingual translators with a background in public health back-translated it into English. The translated version of the questionnaire was evaluated by two bilingual public health professionals working in the field of gerontology. Incomprehensible and ambiguous phrases were identified and substituted with more culturally and linguistically appropriate words and phrases. The Nepali questionnaire was refined after comparing both the initially prepared Nepali version and the back-translated English version. It was finalized after pre-testing it among 10 older people in old age homes and 20 older people in the community living in Kathmandu Valley. The old age homes and wards included in the pre-test were different from the ones in the main study but had similar backgrounds to those of the targeted groups. Data from the pre-test were not included in the final analysis.

Six local research assistants had a background in public health, and they were trained before data collection. They were familiarized with the study objectives, study protocol, research ethics, contents of the questionnaire, the process of data collection, and using tablets for the interview. The principal investigator (RS) and research assistants collected data from April to June 2019. Each survey interview lasted 20–30 minutes.

The number of older people approached for the interview was 950, 250 from old age homes and 700 from the community. Among them, 201 were interviewed in old age homes and 550 in the community. However, data from 57 older people were excluded due to incomplete responses. Finally, data from 694 older people were used for analysis.

## Data analysis

Comparisons tests were performed between the two groups, old age homes and the community, using chi-squared tests for categorical variables. Multiple linear regression analyses were conducted to examine the association of various sociodemographic and health characteristics with frailty. Factors associated with frailty were observed by performing hierarchical linear regression using three models. Model 1 was adjusted for socio-demographic variables (age,

gender, education, marital status, and income). Model 2 included health-related variables (comorbidities, medication use, smoking, and alcohol consumption). Model 3, which is the complete model, was adjusted for residence and satisfaction with home-living environment. Multicollinearity was assessed with variance inflation factor (VIF). Any variable with a VIF of 10 or more was excluded from the analyses. The data were analyzed using STATA/SE 15.1 software (StataCorp, College Station, TX, USA). The level of significance was set to 0.05 (two-tailed).

### Ethical considerations

Ethical approval was obtained from the Research Ethics Committee of The University of Tokyo (2018168NI) and Nepal Health Research Council (160/2019). The older people were ensured of their confidentiality, voluntary participation, and their right to refuse participation at any time. Written informed consent was obtained in signature or thumb imprint from all the older people. Their identity was kept anonymous using identification codes and data is being managed with strict confidentiality. The findings of this study will be shared with concerned stakeholders to bring about positive changes and interventions in older people's care.

## Results

Table 1 summarizes the socio-demographic characteristics of the 694 older people. Among them, 193 resided in old age homes and 501 resided in the community. The mean age of older people in old age homes was 76.8 years (SD 9.9, range 60–104), and 78 (40.4%) were aged 80 and above. Among 193 older people, 131 (67.9%) were women. The number of illiterate older people was 143 (74.1%). Regarding the income level, 170 (88.1%) had an income below 20 USD. Only 30 (15.5%) were dissatisfied with their living environment.

On the other hand, the mean age of older people in the community was 72.6 years (SD 8.2, range 60–100), and 101 (20.1%) were aged 80 and above. Among 501 older people in the community, 256 (51.1%) were women. The number of illiterate older people was 202 (40.3%). Regarding the income, 269 (53.7%) of them had an income below 20 USD. Only 37 (7.4%) were dissatisfied with their living environment. Statistically significant differences were observed in age ($p<0.001$), gender ($p<0.001$), education ($p<0.001$), marital status ($p<0.001$), income ($p<0.001$), and satisfaction with the living environment ($p<0.01$) between older people residing in old age homes and the community.

Table 2 summarizes the health characteristics of the older people. Frailty was more prevalent among older people in old age homes (71.5%) compared to those in the community (56.3%). The total prevalence was 60.5%. Only 21 older people (10.9%) did not suffer from any chronic disease in old age homes and 73 (14.6%) in the community. The number of older people who consumed three or more medicines to manage chronic diseases was 69 (35.8%) in old age homes and 147 (29.4%) in the community. The number of older people who perceived themselves to have a healthy lifestyle was 37 (19.2%) in old age homes compared to 146 (29.1%) in the community. Statistically significant differences were observed in number of comorbidities ($p<0.01$), self-rated lifestyle ($p<0.01$), and alcohol consumption ($p<0.001$) between older people residing in old age homes and the community.

Table 3 illustrates multiple linear regression analyses of the factors associated with frailty. VIF values for all variables were below two, and no multicollinearity was observed. In old age homes, older people who were satisfied with their living environment had lower GFI scores (standardized beta coefficient [β] = -0.20; 95% confidence interval [CI]: -2.95, -0.45). Those who rated their health as fair (β = 0.20; 95% CI = 0.06, 2.42) and unhealthy (β = 0.45; 95% CI = 1.70, 4.38) had higher frailty scores. Similar results were observed in the case of

**Table 1. Sociodemographic characteristics.**

| Variables | Old age homes | | Community | | p-value |
|---|---|---|---|---|---|
| | **n = 193** | | **n = 501** | | |
| | **n** | **%** | **n** | **%** | |
| **Age [Mean (SD)]** | 76.8 (9.9) | | 72.6 (8.2) | | <0.001 |
| 60–69 years | 48 | 24.9 | 185 | 36.9 | |
| 70–79 years | 67 | 34.7 | 215 | 42.9 | |
| > = 80 years | 78 | 40.4 | 101 | 20.2 | |
| **Gender** | | | | | <0.001 |
| Male | 62 | 32.1 | 245 | 48.9 | |
| Female | 131 | 67.9 | 256 | 51.1 | |
| **Education** | | | | | <0.001 |
| Illiterate | 143 | 74.1 | 202 | 40.3 | |
| Non-formal education | 23 | 11.9 | 138 | 27.5 | |
| < Higher secondary | 18 | 9.3 | 96 | 19.2 | |
| > = Higher secondary | 9 | 4.7 | 65 | 13.0 | |
| **Marital Status** | | | | | <0.001 |
| Single | 167 | 86.5 | 252 | 50.3 | |
| Married | 26 | 13.5 | 249 | 49.7 | |
| **Income (in USD)** | | | | | |
| < 20 | 170 | 88.1 | 269 | 53.7 | <0.001 |
| 20 to 100 | 13 | 6.7 | 89 | 17.8 | |
| >100 | 10 | 5.2 | 143 | 28.5 | |
| **Satisfaction with living environment** | | | | | 0.008 |
| No | 30 | 15.5 | 37 | 7.4 | |
| Yes | 163 | 84.5 | 464 | 92.6 | |

USD: US dollars; SD: Standard Deviation.

Chi-square tests.

community-dwelling older people where those who were satisfied with their living environment had lower GFI scores (β = -0.15; 95% CI = -2.62, -0.80) and those who rated their lifestyle as unhealthy had higher GFI scores (β = 0.25; 95% CI = 1.03, 2.53). In addition, older people aged 80 and above had higher GFI scores (β = 0.15; 95% CI = 0.36, 1.86) and those who had achieved education of higher secondary level and above had lower GFI scores (β = -0.13; 95% CI = -2.02, -0.22).

Table 4 shows the hierarchical regression analyses that were run to examine the degree to which the above factors were independently associated with frailty and to show the risk factors associated with it. In old age homes, older people aged 80 or over were more likely to have higher frailty scores in Model 1 (β = 0.18, p < 0.05) which became insignificant in Model 2. Model 3 showed that older people who were satisfied with their living environment (β = -0.20, p < 0.01) were more likely to have lower frailty scores, whereas those who rated their lifestyle as fair (β = 0.20, p < 0.05) and unhealthy (β = 0.45, p < 0.001) were more likely to have higher frailty scores. Model 3 explained a total of 18% of the variance of frailty in old age homes. Self-rated unhealthy lifestyle (β = 0.45, p < 0.001) had the strongest association with frailty.

In the community, older people aged 80 or over were more likely to have higher frailty scores (β = 0.15, p < 0.01) and those with higher secondary education and above (β = -0.13, p < 0.05) were more likely to have the opposite in all the three models. Male older people were more likely to have lower frailty scores (β = -0.12, p < 0.05) than their female counterparts in

**Table 2. Health characteristics.**

| Variables | Old age homes | | Community | | p-value |
|---|---|---|---|---|---|
| | n = 193 | | n = 501 | | |
| | n | % | n | % | |
| **Frailty score (GFI)** | | | | | <0.001 |
| Less than four | 55 | 28.5 | 219 | 43.7 | |
| Four or more | 138 | 71.5 | 282 | 56.3 | |
| **Number of comorbidities** | | | | | 0.005 |
| None | 21 | 10.9 | 73 | 14.6 | |
| One or two | 119 | 61.7 | 236 | 47.1 | |
| Three or more | 53 | 27.4 | 192 | 38.3 | |
| **Number of medicines** | | | | | 0.672 |
| None | 44 | 22.8 | 156 | 31.1 | |
| One or two | 80 | 41.4 | 198 | 39.5 | |
| Three or more | 69 | 35.8 | 147 | 29.4 | |
| **Smoking** | | | | | 0.810 |
| Currently smoking | 35 | 18.1 | 60 | 12 | |
| Past smoker | 47 | 24.4 | 152 | 30.3 | |
| Never | 111 | 57.5 | 289 | 57.7 | |
| **Alcohol consumption** | | | | | <0.001 |
| Yes | 7 | 3.7 | 63 | 12.5 | |
| In the past | 63 | 32.6 | 75 | 15 | |
| Never | 123 | 63.7 | 363 | 72.5 | |
| **Self-rated lifestyle** | | | | | 0.002 |
| Healthy | 37 | 19.2 | 146 | 29.1 | |
| Fair | 94 | 48.7 | 246 | 49.1 | |
| Unhealthy | 62 | 32.1 | 109 | 21.8 | |

Chi-square tests.

Model 1, which became insignificant in the subsequent models. In Model 2, self-rated unhealthy lifestyle ($\beta = 0.27$, $p < 0.001$) increased the likelihood of higher frailty scores. Model 3 showed that older people who were satisfied with their living environment ($\beta = -0.15$, $p < 0.001$) were more likely to have lower frailty scores. Model 3 explained 19.1% of variance of frailty in the community. Self-rated unhealthy lifestyle ($\beta = 0.25$, $p < 0.001$) had the strongest association with frailty followed by satisfaction with the living environment ($\beta = -0.15$, $p < 0.001$).

## Discussion

In this study, frailty score was higher in older people residing in old age homes. Dissatisfaction with the living environment and self-rated unhealthy lifestyle were associated with frailty both in old age homes and in the community. Higher age and higher education were associated with frailty only in the community and not in old age homes.

The percentage of frail older people in this study was higher in those residing in old age homes. This finding is in line with a study in the Netherlands, where older people residing in assisted-living facilities were frailer [6]. The prevalence of frailty among Gurkha welfare pensioners in Nepal was 46.2%, whereas frailty among rural population in eastern part of Nepal was 65% in other study conducted in Nepal [39, 40]. The difference in prevalence of frailty

**Table 3. Multiple linear regression: Factors associated with overall GFI in old age homes (n = 193) and the community (n = 501).**

| Variables | Old age homes | p-value | Community | p-value |
|---|---|---|---|---|
| | β (95% CI) | | β (95% CI) | |
| **Age (vs. 60–69 years)** | | | | |
| 70–79 years | -0.03 (-1.31,0.95) | 0.754 | 0.00 (-0.59, 0.55) | 0.946 |
| > = 80 years | 0.07 (-0.69, 1.54) | 0.454 | 0.15 (0.36, 1.86) | 0.004 |
| **Gender (vs. Female)** | | | | |
| Male | -0.04 (-1.42, 0.86) | 0.630 | -0.05 (-0.91, 0.42) | 0.318 |
| **Education (vs. Illiterate)** | | | | |
| Non-formal education | -0.07 (-2.07, 0.74) | 0.355 | -0.07 (-1.11, 0.12) | 0.116 |
| < Less than higher secondary | 0.02 (-1.29, 1.82) | 0.736 | -0.10 (-1.44, 0.05) | 0.067 |
| Higher secondary and above | 0.04 (-1.78, 2.98) | 0.621 | -0.13 (-2.02, -0.22) | 0.015 |
| **Marital status (vs. Single)** | | | | |
| Married | 0.09 (-0.41, 2.14) | 0.184 | -0.02 (-0.68, 0.42) | 0.649 |
| **Income (in USD) (vs. < 20)** | | | | |
| 20 to 100 | 0.02 (-1.43, 2.00) | 0.744 | -0.01 (-0.79, 0.59) | 0.771 |
| More than 100 | -0.05 (-2.90, 1.44) | 0.509 | -0.02 (-0.78, 0.55) | 0.733 |
| **Number of comorbidities (vs. None)** | | | | |
| 1 or 2 | -0.01 (-1.68, 1.54) | 0.930 | 0.11 (-0.18, 1.52) | 0.124 |
| 3 or more | 0.10 (-1.02, 2.50) | 0.407 | 0.12 (-0.24, 1.74) | 0.139 |
| **Medication use (vs. None)** | | | | |
| 1 to 2 | 0.01 (-1.17, 1.30) | 0.919 | -0.05 (-0.97, 0.39) | 0.404 |
| 3 or more | 0.10 (-0.62, 1.95) | 0.307 | 0.09 (-0.17, 1.39) | 0.126 |
| **Smoking (vs. Never)** | | | | |
| Currently smoking | 0.10 (-0.50, 2.20) | 0.219 | -0.05 (-1.28, 0.30) | 0.223 |
| Past smoker | -0.03 (-1.45, 0.93) | 0.669 | 0.00 (-0.60, 0.55) | 0.929 |
| **Alcohol consumption (vs. Never)** | | | | |
| In the past | 0.00 (-1.06, 1.02) | 0.967 | -0.03 (-0.97, 0.46) | 0.485 |
| Yes | -0.02 (-2.73, 1.95) | 0.741 | -0.07 (-1.40, 0.14) | 0.111 |
| **Self-rated lifestyle (vs. Healthy)** | | | | |
| Fair | 0.20 (0.06, 2.42) | 0.040 | 0.08 (-0.10, 1.08) | 0.101 |
| Unhealthy | 0.45 (1.70, 4.38) | <0.001 | 0.25 (1.03, 2.53) | <0.001 |
| **Satisfaction with living environment (vs. No)** | | | | |
| Yes | -0.20 (-2.95, -0.45) | 0.008 | -0.15 (-2.62, -0.80) | <0.001 |

β: Standardized beta coefficient.

CI: Confidence interval.

could be because of the use of different frailty instruments used in these studies. The higher frailty score in old age homes in this study could be because of higher rates of self-rated unhealthy lifestyle and dissatisfaction among their residents than community-dwelling older people which are discussed below. It could also be because of lack of family support as suggested by a study conducted in Nepal [40].

Older people who were satisfied with their living environment were less likely to be frail both in old age homes and in the community. This finding was previously found in European studies but is new for a low-income setting [41, 42]. Frailty state can deteriorate when older people lack resources to support aging in their living environment and cannot maintain independent living [43, 44]. A living environment that is not age-friendly could lead to limited physical activity and reduced opportunities, thus contributing to loneliness and isolation,

**Table 4. Hierarchical regression analyses: Factors associated with overall GFI in old age homes (n = 193) and the community (n = 501).**

| Variables | Old age homes | | | Community | | |
|---|---|---|---|---|---|---|
| | **Model 1** | **Model 2** | **Model 3** | **Model 1** | **Model 2** | **Model 3** |
| **Age (vs. 60–69 years)** | | | | | | |
| 70–79 years | 0.06 | 0.03 | -0.03 | 0.03 | -0.01 | 0.00 |
| $\geq$ 80 years | 0.18* | 0.07 | 0.07 | 0.19* | 0.14* | 0.15** |
| **Gender (vs. Female)** | | | | | | |
| Male | 0.01 | -0.02 | -0.04 | -0.12* | -0.05 | -0.05 |
| **Education (vs. Illiterate)** | | | | | | |
| Non-formal education | -0.02 | -0.07 | -0.07 | -0.06 | -0.08 | -0.07 |
| < Higher secondary | 0.04 | 0.03 | 0.02 | -0.08 | -0.10* | -0.10 |
| $\geq$ Higher secondary | 0.00 | 0.04 | 0.04 | -0.12* | -0.12* | -0.13* |
| **Marital status (vs. Single)** | | | | | | |
| Married | 0.10 | 0.08 | 0.09 | -0.01 | -0.03 | -0.02 |
| **Income (in USD) (vs. Less than 20)** | | | | | | |
| 20 to 100 | 0.05 | 0.01 | 0.02 | -0.06 | -0.02 | -0.01 |
| > 100 | -0.05 | -0.04 | -0.05 | -0.05 | -0.03 | -0.02 |
| **Number of comorbidities (vs. None)** | | | | | | |
| 1 or 2 | | 0.00 | -0.01 | | 0.10 | 0.11 |
| 3 or more | | 0.11 | 0.10 | | 0.12 | 0.12 |
| **Medication use (vs. None)** | | | | | | |
| 1 to 2 | | 0.00 | 0.01 | | -0.04 | -0.05 |
| 3 or more | | 0.10 | 0.10 | | 0.10 | 0.09 |
| **Smoking (vs. Never)** | | | | | | |
| Currently smoking | | 0.15 | 0.10 | | -0.06 | -0.05 |
| Past smoker | | -0.04 | -0.03 | | 0.01 | 0.00 |
| **Alcohol consumption (vs. Never)** | | | | | | |
| In the past | | -0.01 | 0.00 | | -0.02 | -0.03 |
| Yes | | -0.03 | -0.02 | | -0.06 | -0.07 |
| **Self-rated lifestyle (vs. Healthy)** | | | | | | |
| Fair | | 0.20* | 0.20* | | 0.10 | 0.08 |
| Unhealthy | | 0.52*** | 0.45*** | | 0.27*** | 0.25*** |
| **Satisfaction with living environment (vs. No)** | | | | | | |
| Yes | | | -0.20** | | | -0.15*** |
| **$R^2$ (%)** | 3.8 | 23.5 | 26.5 | 9.0 | 20.2 | 22.3 |
| **$\Delta R^2$ (%)** | -0.9 | 15.1*** | 18.0*** | 7.4*** | 17.1*** | 19.1*** |

Values are presented as standardized beta coefficients (β).

Statistical significance indicated by

*$p < 0.05$

**$p < 0.01$, and

***$p < 0.001$.

$R^2$: variance; $\Delta R^2$: change in variance.

leading to decline in cognitive functions [45]. Nepal lacks standards for operating old age homes, including age-friendly building standards and studies on this topic are scarce [30, 46]. Older people were dissatisfied with the lack of rampways and elevators, and traditional squat toilets, in a hospital-setting in Nepal [47]. Old age home residents complained of having to climb stairs, dirty restrooms, inaccessible bathrooms, lacking railing support, and inaccessible

doorways for wheelchair users [23, 30]. They also complained of rude behavior from caretaking staff and inadequate staff in old age home [23]. Factors that can promote satisfaction with the living environment in France and the Netherlands were the availability of basic living facilities located on the floor of residence, enough opportunities for social contact, and feeling of safety at home and the neighborhood [48, 49].

In this study, older people who perceived themselves to have an unhealthy lifestyle were more likely to be frail in both settings. This finding is also new for a low-income country as previous studies analyzed lifestyle factors such as alcohol and smoking but not self-rated lifestyle [21, 41, 42]. Self-rated health was independently associated with frailty in a study conducted in Mexico [50]. Older people who perceive themselves to be unhealthy usually are not physically active, do not consume a healthy diet, and lead a sedentary lifestyle which may accelerate the development of frailty [44, 51]. This finding is consistent with previous studies in the Netherlands and Hong Kong, which show that lifestyle factors such as diet, physical activity, smoking, and alcohol consumption have a crucial effect on frailty [9, 10].

Frailty score increased with advancing age in community-dwelling older people. Age is a known risk factor of frailty. As people age, oxidative damage results in cell death, necrosis, and proliferation because of damage to DNA, which ultimately leads to loss of muscle mass called sarcopenia [52]. Sarcopenia is directly related to age and manifests as weak muscle strength, slowed gait, and poor balance [52]. Physiological and functional deficits increase with age which lead to frailty [4].

Frailty score was lower in educated community-dwelling older people, which is consistent with findings from studies conducted in Italy and the Netherlands [53, 54]. Education capacitates older people to identify their health problems and seek healthcare when needed through health-related knowledge and behavior [55]. It also improves their cognitive performance and decreases functional limitations [56]. Higher education enables older people to access health information, communicate better, and perform complex activities at home and in the community. Education may affect the prospects of finding a well-paying job past the retirement age and, in turn, prevent the decline of function through physical, cognitive, and psychological activation [57]. Health education and promotion activities in older people have shown to slow down frailty in Japan [58].

This study has some limitations. Due to the cross-sectional design, cause-effect interpretations between frailty and adverse life outcomes could not be established [6, 59]. Reverse causality is possible between satisfaction with the living environment, self-rated health, and frailty. A longitudinal design could have overcome these limitations. However, the objective of this study was to assess the association between exposure and outcome variables rather than to observe a cause-effect relationship. In addition, the findings may not be generalizable to the entire population of Nepal because the study was conducted in an urban setting. This study, however, recruited older people from various socioeconomic backgrounds to overcome this limitation. Despite these limitations, this is one of the few studies to measure the frailty status and explore the factors associated with it in both old age homes and the community in a low-income setting.

## Conclusion

This study found dissatisfaction with the living environment and self-rated unhealthy lifestyle as the factors associated with frailty in both old age homes and the community. In addition, old age and lower education levels were also associated with frailty in community-dwelling older people. The results call for provisions to prevent frailty by focusing on the modifiable risk factors. Reforms in building standards are a dire need to promote an age-friendly

community. The living spaces should be created with the needs of older people in consideration with facilities for physical activity and social connections, both at home and in the neighborhood. Similarly, improvement in lifestyle is possible through simple resistance exercises and addition of nutritional supplements in diet to increase the lean body mass, and improve strength and walking speed. Also, the management of frailty should include health education through disseminating information and increasing awareness on frailty.

## Acknowledgments

We would like to express our gratitude to all the older people who participated in this study. We would also like to thank all organizations, especially Ageing Nepal, for their collaboration and efforts.

## Author Contributions

**Conceptualization:** Richa Shah, Masamine Jimba.

**Data curation:** Richa Shah, Rogie Royce Carandang, Akira Shibanuma, Masamine Jimba.

**Formal analysis:** Richa Shah, Rogie Royce Carandang, Akira Shibanuma.

**Funding acquisition:** Richa Shah.

**Investigation:** Richa Shah, Akira Shibanuma, Masamine Jimba.

**Methodology:** Richa Shah, Rogie Royce Carandang, Ken Ing Cherng Ong, Junko Kiriya, Masamine Jimba.

**Project administration:** Richa Shah, Masamine Jimba.

**Resources:** Richa Shah, Rogie Royce Carandang, Akira Shibanuma, Ken Ing Cherng Ong, Masamine Jimba.

**Software:** Richa Shah, Akira Shibanuma.

**Supervision:** Rogie Royce Carandang, Akira Shibanuma, Junko Kiriya, Masamine Jimba.

**Validation:** Richa Shah, Masamine Jimba.

**Visualization:** Richa Shah, Rogie Royce Carandang, Ken Ing Cherng Ong.

**Writing – original draft:** Richa Shah.

**Writing – review & editing:** Richa Shah, Rogie Royce Carandang, Akira Shibanuma, Ken Ing Cherng Ong, Junko Kiriya, Masamine Jimba.

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
