## [Decision Letter · Decision Letter 0]

22 Feb 2021

PONE-D-20-40524

Understanding frailty among older people living in old age homes and the community in Nepal: a cross-sectional study

PLOS ONE

Dear Dr. Shah,

Thank you for submitting your manuscript to PLOS ONE. After careful consideration, we feel that it has merit but does not fully meet PLOS ONE’s publication criteria as it currently stands. Therefore, we invite you to submit a revised version of the manuscript that addresses the points raised during the review process.

We look forward to receiving your revised manuscript.

Kind regards,

Toshiyuki Ojima, M.D., Ph.D

Academic Editor

PLOS ONE

Additional Editor Comments:

Please explain sampling procedure in methods section.

Reviewers' comments:

Reviewer's Responses to Questions

**Comments to the Author**

1. Is the manuscript technically sound, and do the data support the conclusions?

Reviewer #1: Yes

Reviewer #2: Yes

2. Has the statistical analysis been performed appropriately and rigorously? 

Reviewer #1: Yes

Reviewer #2: Yes

3. Have the authors made all data underlying the findings in their manuscript fully available?

Reviewer #1: No

Reviewer #2: Yes

4. Is the manuscript presented in an intelligible fashion and written in standard English?

Reviewer #1: Yes

Reviewer #2: No

5. Review Comments to the Author

Reviewer #1: Page 7, line 117: I think it will be beneficial if authors provide explanation of how was moderate to severe cognitive impairment was assessed.

Page 9, line 178: Please mention how many old age homes were included in the study.

Page 10. Ethical considerations: As severe frailty indicates higher vulnerability to adverse health outcomes such as falls and disability in older people, what sort of interventions were provided to prevent such outcomes in this study? (As authors have mentioned that frailty can also be reversed with appropriate interventions). I think from an ethical standpoint, authors need to provide clarification on this aspect.

Tables 3 and 4: In the variable Gender, is the reference category male or female?

Discussion: Proportion of older people with severe frailty was higher in old age homes than community. Authors could discuss the current policy provisions for old age homes in Nepal if available, and suggestions to improve, some of which the authors have already touched upon in Page 17.

Reviewer #2: Congratulations authors for this study "…Fragility of among older people…" another dimensions of older people's study from Nepal. Your study is important. Here are my few questions/queries related to your study.

1. Introduction: Proportion of older people in Nepal in 2011 census is 8.1%, I suggest using the exact data. Please refer the article for detail statistics: Chalise HN. Provincial situation of elderly population in Nepal. Am J Aging Sci Res 2020; 1(1): 9-11.

Your citation 23 and references does not seems relevant, use the relevant citations as there are many studies related with issues of older people.

2. Methodology: Your methodological part needs further improvement. Please re-write this part including, how you determined the sample size, how did you measured the cognitive impairment and other also (inclusion and exclusion), how and where did you do the community sampling and at what basis? Please write it systematically, also read other articles published in PlosOne.

You have used Groningen Frailty Indicator (GFI) to assess the fraility of Nepalese older people. How did you assess the reliability and validity of this tool in the Nepalese culture?

3. "Table 1 summarizes the socio-demographic characteristics with of the 694 older people and 210 frailty scores.", I did not find the frailty score in table 1.

4. Discussion:

Please discuss your findings including other relevant studies from Nepal.

You did not discuss about the prevalence of frailty based on your research, why? It may help to know the situation of Nepal where it is.

Please also discuss about some research related with functional disability of Nepalese older people (Functional Disability on Instrumental/Activities of Daily Livings among Rural Older People in Nepal) somewhere in your article as this is one of the important variable for the frailty. Previous article from Nepal shows that functional disability is very high among Nepalese older people. High functional disability means high frailty.

Your study shows old age home residents have high frailty score, what may be the reason. Does it mean living in old age home results high frailty score.

All the best !

6. PLOS authors have the option to publish the peer review history of their article (what does this mean?). If published, this will include your full peer review and any attached files.

Reviewer #1: **Yes: **Pranil Man Singh Pradhan, MD

Reviewer #2: No

---

## [Author Response · Author response to Decision Letter 0]

30 Mar 2021

Dear Editor and Reviewers,

Thank you very much for your kind and supportive comments. Please kindly find the following responses. All the responses described below are also seen in "Responses to Reviewers.docx" attached.

Editor’s comment: 

1. Thank you for submitting your manuscript to PLOS ONE. After careful consideration, we feel that it has merit but does not fully meet PLOS ONE’s publication criteria as it currently stands. Therefore, we invite you to submit a revised version of the manuscript that addresses the points raised during the review process.

Response to 1: Thank you very much for your kind and supportive comment.

Response to 2: We have updated the manuscript according to PLOS ONE’s style requirements. 

3. Please explain sampling procedure in methods section. 

Response to 3: We have explained the sampling procedure in the revised manuscript. We used convenience sampling to select the older people in this study. Please refer to page 6, lines 115-129. 

“Convenience sampling was used to select the older people in the community as data on the number of older people aged only over 70 was available. The heads of 20 wards offices from Kathmandu, six from Lalitpur, and five from Bhaktapur districts were approached with a request letter for permission to conduct the study from The University of Tokyo. These wards were selected purposively based on the ease of access. A ward is the smallest local unit or political division used for electoral purposes [35]. In Kathmandu district, verbal permission was obtained from six wards, in Lalitpur district from three wards, and three wards in Bhaktapur district. Local leaders from the selected wards assisted in approaching older people. Older people were recruited through home visits based on the inclusion and exclusion criteria. 

An official list of old age homes in Nepal was not available. A non-governmental organization called Ageing Nepal, which works for the welfare of the older people in Nepal, provided its own list of old age homes in Nepal. The heads of 15 old age homes were approached first via telephone calls followed by face-to-face meetings with a request letter for permission from The University of Tokyo. Among them, ten provided verbal permission after which interviews were conducted.” 

4. In your Data Availability statement, you have not specified where the minimal data set underlying the results described in your manuscript can be found. PLOS defines a study's minimal data set as the underlying data used to reach the conclusions drawn in the manuscript and any additional data required to replicate the reported study findings in their entirety. All PLOS journals require that the minimal data set be made fully available. For more information about our data policy, please see http://journals.plos.org/plosone/s/data-availability. Upon re-submitting your revised manuscript, please upload your study’s minimal underlying data set as either Supporting Information files or to a stable, public repository and include the relevant URLs, DOIs, or accession numbers within your revised cover letter. For a list of acceptable repositories, please see http://journals.plos.org/plosone/s/data-availability#loc-recommended-repositories. Any potentially identifying patient information must be fully anonymized.

Response to 4: We have shared the data set on Figshare repository, which is available at “https://figshare.com/s/da4e349dc1f2597ed2cf”.

Reviewer #1

1. Page 7, line 117: I think it will be beneficial if authors provide explanation of how was moderate to severe cognitive impairment was assessed. 

Response to 1: Based on the medical records of older people, we excluded those who had clinically diagnosed moderate to severe cognitive impairment. We also excluded older people who had obvious signs of cognitive impairment that affected their ability to answer correctly to the survey questions. These signs included inability to remember their name or location of their residence. Please refer to page 7, lines 131-140. 

“At old age homes, medical records were available. Based on the records, older people were excluded from this study if they suffered from moderate to severe cognitive impairment, severe hearing impairment, muteness, and mental illnesses such as psychosis, bipolar disorder, and schizophrenia. Most community-dwelling older people also had their medical records which the interviewers could access. In the absence of medical records or inadequate information in them, the older people were excluded if they had visible signs of cognitive impairment such as inability to remember their name or the location of their residence, which affected their ability to comprehend the study procedure and to cooperate with the investigators.” 

2. Page 9, line 178: Please mention how many old age homes were included in the study. 

Response to 2: In this study, we included ten old age homes. We have revised the section on pages 6-7, lines 126-129 to clarify the number of old age homes included in this study. 

“The heads of 15 old age homes were approached first via telephone calls followed by face-to-face meetings with a request letter for permission from The University of Tokyo. Among them, ten provided verbal permission after which interviews were conducted.” 

3. Page 10. Ethical considerations: As severe frailty indicates higher vulnerability to adverse health outcomes such as falls and disability in older people, what sort of interventions were provided to prevent such outcomes in this study? (As authors have mentioned that frailty can also be reversed with appropriate interventions). I think from an ethical standpoint, authors need to provide clarification on this aspect. 

Response to 3: The findings will be shared with Kathmandu Metropolitan Government office and Ageing Nepal, a leading organization working towards well-being of Nepaleseolder people, after the publication of this study. Please refer to page 11, lines 217-218. 

“The findings of this study will be shared with concerned stakeholders to bring about positive changes and interventions in older people’s care.” 

4. Tables 3 and 4: In the variable Gender, is the reference category male or female? 

Response to 4: We have added the reference category (female) in tables 3 and 4 on pages 14 and 16, respectively. 

5. Discussion: Proportion of older people with severe frailty was higher in old age homes than community. Authors could discuss the current policy provisions for old age homes in Nepal if available, and suggestions to improve, some of which the authors have already touched upon in Page 17. 

Response to 5: We did not find any policy provisions for old age homes in Nepal. Most old age homes are running without a standard protocol. We have added a sentence in the discussion section highlighting this. Please refer to page 18, lines 300-301. 

“Nepal lacks standards for operating old age homes, including building standards that are age-friendly and studies on this topic are scarce [30, 46].”

Reviewer #2: 

1. Congratulations authors for this study "…Fragility of among older people…" another dimensions of older people's study from Nepal. Your study is important.

Response to 1: Thank you very much for your kind and supportive comments.

2. Introduction: Proportion of older people in Nepal in 2011 census is 8.1%, I suggest using the exact data. Please refer the article for detail statistics: Chalise HN. Provincial situation of elderly population in Nepal. Am J Aging Sci Res 2020; 1(1): 9-11. 

Response to 2: We have revised the percentage of older people in Nepal to 8.1% in the introduction section. Please refer to page 4, lines 71-72. 

“According to the latest census in 2011, the population of individuals aged 60 years old and above was 2.1 million in Nepal, which was 8.1% of the total population [27].” 

3. Your citation 23 and references does not seems relevant, use the relevant citations as there are many studies related with issues of older people. 

Response to 3: We have revised citation 23. We removed the citation “Shrestha DRS, A.; Ghimire, J. Emerging challenges in family planning programme in Nepal. J Nepal Health Res Counc. 2012;10(21):108-12.” and added two relevant citations on page 4, line 67. 

“The number of older people in Nepal is rising rapidly as elsewhere in the world [23, 24]. 

23. Mishra S, Chalise HN. Health status of elderly living in government and private old age home in Nepal. Sciences. 2018;11(4):173-8. 

24.Chalise HN. Demographic situation of population ageing in Nepal. Kathmandu Univ Med J (KUMJ). 2006;4(3):354-62.” 

4. Methodology: Your methodological part needs further improvement. Please re-write this part including, how you determined the sample size, how did you measured the cognitive impairment and other also (inclusion and exclusion), how and where did you do the community sampling and at what basis? Please write it systematically, also read other articles published in PlosOne.

Response to 4: We revised the methods section based on PLOS ONE’s guidelines. We also included a section on sample size calculation and community sampling, and elaborated on the measurement of cognitive impairment. 

Page 6, lines 108-114 

Sample size: “The sample size was calculated based on a previous study by Lin et al. which divided the population into two groups, frail and robust with means of 52.6 (SD [Standard deviation] 8.8) for frail and 56.2 for non-frail (SD 12.8) [34]. OpenEpi version 3.01 was used for calculation with 80% power, 5% significance level, and 95% confidenceinterval. Thus, the minimum sample size obtained was 115 for old age homes and 348 for the community. However, anticipating about 30% refusal rate and incomplete responses, the final minimum sample size was set to 164 for old age homes and 497 for the community.” 

Page 6, lines 115-129 

Sampling: “Convenience sampling was used to select the older people in the community as data on the number of older people aged only over 70 was available. The heads of 20 wards offices from Kathmandu, six from Lalitpur, and five from Bhaktapur districts were approached with a request letter for permission to conduct the study from The University of Tokyo. These wards were selected purposively based on the ease of access. A ward is the smallest local unit or political division used for electoral purposes [35]. In Kathmandu district, verbal permission was obtained from six wards, in Lalitpur district from three wards, and three wards in Bhaktapur district. Local leaders from the selected wards assisted in approaching older people. Older people were recruited through home visits based on the inclusion and exclusion criteria. 

An official list of old age homes in Nepal was not available. A non-governmental organization called Ageing Nepal, which works for the welfare of the older people in Nepal, provided its own list of old age homes in Nepal. The heads of 15 old age homes were approached first via telephone calls followed by face-to-face meetings with a request letter for permission from The University of Tokyo. Among them, ten provided verbal permission after which interviews were conducted.” 

Page 7, lines 131-146 

Inclusion and exclusion criteria including cognitive impairment: “The older people included in this study were aged 60 years and over and residing in the community and old age homes in Kathmandu Valley at the time of data collection [23, 30]. At old age homes, medical records were available. Based on the records, older people were excluded from this study if they suffered from moderate to severe cognitive impairment, severe hearing impairment, muteness, and mental illnesses such as psychosis, bipolar disorder, and schizophrenia. Most community-dwelling older people also had their medical records which the interviewers could access. In the absence of medical records or inadequate information in them, the older people were excluded from the study if they had visible signs of cognitive impairment such as inability to remember their name or the location of their residence, which affected the ability to comprehend the study procedure and to cooperate with the investigators. Sensory and cognitive decline may occur in old age, resulting in loss of memory, impaired comprehension and judgment, or inability to respond accurately. They could incapacitate older people from consenting to participate in the study or providing precise information. Only one older person was recruited from each household based on inclusion and exclusion criteria in the community. If a couple resided together in an old age home, only one of them was recruited. Based on the above, 35 older people were excluded from this study.” 

5. You have used Groningen Frailty Indicator (GFI) to assess the frailty of Nepalese older people. How did you assess the reliability and validity of this tool in the Nepalese culture? 

Response to 5: The internal consistency reliability of GFI was measured using Cronbach’s alpha. The instrument was checked for face and content validity during translation and back-translation of the questionnaire as well as during pre-testing by bilingual medical doctors with a background in public health and a researcher in the field of gerontology. 

Page 8, lines 163-165 

“The internal consistency of GFI in this study was measured using Cronbach’s alpha which was 0.74 for old age homes, 0.76 for the community, and 0.75 for combined, indicating high reliability [38].” 

Page 9. Lines 177-188 “The questionnaire was initially prepared in English and translated into the Nepali language by a bilingual medical doctor and public health professional. Two independent bilingual translators with a background in public health back-translated it into English. The translated version of the questionnaire was evaluated by two bilingual public health professionals working in the field of gerontology. Incomprehensible and ambiguous phrases were identified and substituted with more culturally and linguistically appropriate words and phrases. The Nepali questionnaire was refined after comparing both the initially prepared Nepali version and the back-translated English version. It was finalized after pre-testing it among 10 older people in old age homes and 20 older people in the community living in Kathmandu Valley. The old age homes and wards included in the pre-test were different from the ones in the main study but had similar backgrounds to those of the targeted groups. Data from pre-test were not included in the final analysis.” 

6. "Table 1 summarizes the socio-demographic characteristics with of the 694 older people and 210 frailty scores.", I did not find the frailty score in table 1. 

Response to 6: Thank you for bringing our attention to this error. We have corrected the sentence to “Table 1 summarizes the socio-demographic characteristics of the 694 older people” on page 11, line 220. Frailty scores are given in Table 2 on page 12. 

7. Please discuss your findings including other relevant studies from Nepal. You did not discuss about the prevalence of frailty based on your research, why? It may help to know the situation of Nepal where it is. Your study shows old age home residents have high frailty score, what may be the reason. Does it mean living in old age home results high frailty score. 

Response to 7: In our revised manuscript, we added a description of prevalence of frailty in this study and compared it with other studies conducted in Nepal in the discussion section. We also discussed about high frailty scores in old age homes which could be a result of higher rates of self-rated unhealthy lifestyle and dissatisfaction among their residents compared to community-dwelling older people. Please refer to page 17, lines 280-293. 

“In this study, frailty score was higher in older people residing in old age homes. Dissatisfaction with the living environment and self-rated unhealthy lifestyle were associated with frailty both in old age homes and in the community. Higher age and higher education were associated with frailty only in the community and not in old age homes. 

The percentage of frail older people in this study was higher in those residing in old age homes. This finding is in line with a study in the Netherlands, where older people residing in assisted-living facilities were frailer [6]. The prevalence of frailty among Gurkha welfare pensioners in Nepal was 46.2%, whereas frailty among rural population in eastern part of Nepal was 65% in other study conducted in Nepal [39, 40]. The difference in prevalence of frailty could be because of the use of different frailty instruments used in these studies. The higher frailty score in old age homes in this study could be because of higher rates of self-rated unhealthy lifestyle and dissatisfaction among their residents compared to community-dwelling older people which are discussed below. It could also be as a result of lack of family support as suggested by a study conducted in Nepal [40].” 

8. Please also discuss about some research related with functional disability of Nepalese older people (Functional Disability on Instrumental/Activities of Daily Livings among Rural Older People in Nepal) somewhere in your article as this is one of the important variable for the frailty. Previous article from Nepal shows that functional disability is very high among Nepalese older people. High functional disability means high frailty. 

Response to 8: We have mentioned in the introduction section that frailty can lead to adverse health outcomes such as disability. We did not explain it further as we did not assess the association between frailty and disability in this study.

---

## [Decision Letter · Decision Letter 1]

19 Apr 2021

Understanding frailty among older people living in old age homes and the community in Nepal: A cross-sectional study

PONE-D-20-40524R1

Dear Dr. Shah,

We’re pleased to inform you that your manuscript has been judged scientifically suitable for publication and will be formally accepted for publication once it meets all outstanding technical requirements.

Kind regards,

Toshiyuki Ojima, M.D., Ph.D

Academic Editor

PLOS ONE

Additional Editor Comments (optional):

Reviewers' comments:

Reviewer's Responses to Questions

**Comments to the Author**

1. If the authors have adequately addressed your comments raised in a previous round of review and you feel that this manuscript is now acceptable for publication, you may indicate that here to bypass the “Comments to the Author” section, enter your conflict of interest statement in the “Confidential to Editor” section, and submit your "Accept" recommendation.

Reviewer #1: All comments have been addressed

Reviewer #2: All comments have been addressed

2. Is the manuscript technically sound, and do the data support the conclusions?

Reviewer #1: Yes

Reviewer #2: Yes

3. Has the statistical analysis been performed appropriately and rigorously? 

Reviewer #1: Yes

Reviewer #2: Yes

4. Have the authors made all data underlying the findings in their manuscript fully available?

Reviewer #1: Yes

Reviewer #2: No

5. Is the manuscript presented in an intelligible fashion and written in standard English?

Reviewer #1: Yes

Reviewer #2: Yes

6. Review Comments to the Author

Reviewer #1: (No Response)

Reviewer #2: Thank you author for addressing the majority of my comments/suggestions and queries. Wish you all the best.

7. PLOS authors have the option to publish the peer review history of their article (what does this mean?). If published, this will include your full peer review and any attached files.

Reviewer #1: **Yes: **Pranil Man Singh Pradhan

Reviewer #2: **Yes: **Hom Nath Chalise, PhD

---

## [Editor Report · Acceptance letter]

21 Apr 2021

PONE-D-20-40524R1 

Understanding frailty among older people living in old age homes and the community in Nepal: A cross-sectional study 

Dear Dr. Shah:

I'm pleased to inform you that your manuscript has been deemed suitable for publication in PLOS ONE. Congratulations! Your manuscript is now with our production department. 

Kind regards, 

on behalf of

Dr. Toshiyuki Ojima 

Academic Editor

PLOS ONE